# Dexrazoxane does not mitigate early vascular toxicity induced by doxorubicin in mice

**Matthias Bosman**[ID][1]*, **Dustin N. Krüger**[1], **Kasper Favere**[1,2,3], **Guido R. Y. De Meyer**[ID][1], **Constantijn Franssen**[2,3], **Emeline M. Van Craenenbroeck**[2,3], **Pieter-Jan Guns**[ID][1]

1 Laboratory of Physiopharmacology, Faculty of Medicine and Health Sciences, Faculty of Pharmaceutical, Biomedical and Veterinary Sciences, Campus Drie Eiken, University of Antwerp, Antwerp, Belgium,
2 Research Group Cardiovascular Diseases, GENCOR, University of Antwerp, Antwerp, Belgium,
3 Department of Cardiology, Antwerp University Hospital (UZA), Edegem, Belgium

* matthias.bosman@uantwerpen.be

## Abstract

Apart from cardiotoxicity, the chemotherapeutic agent doxorubicin (DOX) provokes acute and long-term vascular toxicity. Dexrazoxane (DEXRA) is an effective drug for treatment of DOX-induced cardiotoxicity, yet it remains currently unknown whether DEXRA prevents vascular toxicity associated with DOX. Accordingly, the present study aimed to evaluate the protective potential of DEXRA against DOX-related vascular toxicity in a previously-established *in vivo* and *ex vivo* model of vascular dysfunction induced by 16 hour (h) DOX exposure. Vascular function was evaluated in the thoracic aorta in organ baths, 16h after administration of DOX (4 mg/kg) or DOX with DEXRA (40 mg/kg) to male C57BL6/J mice. In parallel, vascular reactivity was evaluated after *ex vivo* incubation (16h) of murine aortic segments with DOX (1 µM) or DOX with DEXRA (10 µM). In both *in vivo* and *ex vivo* experiments, DOX impaired acetylcholine-stimulated endothelium-dependent vasodilation. In the *ex vivo* setting, DOX additionally attenuated phenylephrine-elicited vascular smooth muscle cell (VSMC) contraction. Importantly, DEXRA failed to prevent DOX-induced endothelial dysfunction and hypocontraction. Furthermore, RT-qPCR and Western blotting showed that DOX decreased the protein levels of topoisomerase-IIβ (TOP-IIβ), a key target of DEXRA, in the heart, but not in the aorta. Additionally, the effect of N-acetylcysteine (NAC, 10 µM), a reactive oxygen species (ROS) scavenger, was evaluated *ex vivo*. NAC did not prevent DOX-induced impairment of acetylcholine-stimulated vasodilation. In conclusion, our results show that DEXRA fails to prevent vascular toxicity resulting from 16h DOX treatment. This may relate to DOX provoking vascular toxicity in a ROS- and TOP-IIβ-independent way, at least in the evaluated acute setting. However, it is important to mention that these findings only apply to the acute (16h) treatment period, and further research is warranted to delineate the therapeutic potential of DEXRA against vascular toxicity associated with longer-term repetitive DOX dosing.

**Data Availability Statement:** All relevant data are within the paper and its Supporting Information files.

**Funding:** This work was supported by the Fund for Scientific Research (FWO; https://www.fwo.be/) Flanders to whom M.B. and K.F. are predoctoral fellows [grant number: 1S33720N and 11C6321N, respectively]. E.V.C. is holder of a senior clinical investigator grant from FWO Flanders [grant number: 1804320N]. C.F. is holder of a clinical mandate from Foundation against Cancer (2021-034; https://www.kanker.be/). Furthermore, the research is supported by a DOCPRO4 grant of the Research Council of the University of Antwerp (ID: 39984; https://www.uantwerpen.be/nl/onderzoek/beleid/financiering-onderzoek/interne-financiering/bof/) and by the INSPIRE project, which has received funding from the European Union's Horizon 2020 Research and Innovation Program (H2020-MSCA-ITN program, Grant Agreement: No858070; https://research-and-innovation.ec.europa.eu/funding/funding-opportunities/funding-programmes-and-open-calls/horizon-2020_en). The funders had no role in study design, data collection and analysis, decision to publish, or preparation of the manuscript.

**Competing interests:** The authors have declared that no competing interests exist.

## Introduction

Despite cardiotoxic side effects, the anthracycline doxorubicin (DOX) remains a first-line chemotherapeutic agent, especially in breast cancer and lymphoma [1, 2]. Apart from compromising cardiac function, there is compelling clinical and preclinical evidence that DOX adversely impacts the vasculature. More specifically, DOX-treated childhood cancer survivors exhibit a higher incidence of systemic hypertension and coronary artery disease [3]. Moreover, DOX has been shown to impair endothelial function and increase arterial stiffness during therapy [4–12], which have been associated with future cardiovascular disease in the general population [13–18]. In this respect, it has been suggested that early vascular toxicity during DOX therapy contributes to cardiac disease in cancer survivors later in life [3].

At present, dexrazoxane (DEXRA), an iron chelator, is the only available drug for treating DOX-induced cardiotoxicity in patients [19]. Several mechanisms have been proposed through which DOX provokes cardiotoxicity, including targeting of the DNA-repair enzyme topoisomerase-IIβ (TOP-IIβ) and formation of reactive oxygen species (ROS) [20–23]. DEXRA, on the other hand, prevents DOX-related cardiotoxicity by depleting TOP-IIβ [20, 21]. In addition, DEXRA attenuates DOX-induced ROS formation in the heart [24], although its antioxidant role has recently been challenged [20]. While the cardioprotective role of DEXRA against DOX-induced cardiotoxicity has been well-established, it remains currently unknown whether DEXRA prevents vascular toxicity associated with DOX. As such, the present study aimed to evaluate the capability of DEXRA to mitigate DOX-related vascular toxicity in a previously-established *in vivo* and *ex vivo* model of impaired vascular function induced by short-term (16 hours (h)) DOX exposure.

## Materials & methods

### Animals and ethical approval

Male *C57BL/6J* mice (age: 10–12 weeks; body weight: 24–30 g; Charles River) were housed in the animal facility of the University of Antwerp in standard cages with 12–12 h light–dark cycles with access to water *ad libitum* and regular chow, containing in %: crude protein, 19.0; crude fat, 3.3; crude fibre, 5.0; crude ash, 6.4; starch, 35.9; sugar, 5.4 and nitrogen-free extracts, 54.6. All experiments were approved by the Animal Ethics Committee of the University of Antwerp (file 2021–19) and conformed with the ARRIVE guidelines and with the Belgian Royal Decree of 2013.

### DOX treatment

**In vitro.** Cardiomyocytes (passage 3–4), endothelial cells (ECs; passage 1–2) and vascular smooth muscle cells (VSMCs; passage 1–2), all derived from wild-type *C57BL/6J* mice, were cultured in Dulbecco's modified Eagle medium (DMEM) medium (Thermo Fisher Scientific) supplemented with 1% penicillin/streptomycin and 10% foetal bovine serum (Thermo Fisher Scientific). Cardiomyocytes, ECs and VSMCs were treated with either phosphate-buffered saline solution (PBS) as vehicle (1:100), DOX (1 μM), DEXRA (10 μM) or a combination of DOX (1 μM) and DEXRA (10 μM) for 16 hours (h). The same treatment period (16h) was used for the *in vivo* and *ex vivo* experiments (see further) as we have previously shown that, at this time point, DOX provokes vascular dysfunction *in vivo* and *ex vivo* [25], thus representing a readily-accessible model of acute DOX-induced vascular toxicity. The DOX dose of 1 μM was chosen to resemble plasma concentrations in patients [26]. A DEXRA dose of 10 μM was used since, in patients, a DEXRA to DOX ratio of 10 to 1 is recommended [27]. DEXRA was

administered 15 minutes prior to DOX addition to allow the uptake of DEXRA first. After 16h, cell viability was evaluated with a Countess™ II FL instrument (Thermo Fisher Scientific).

**In vivo.** DOX (4 mg/kg), a combination of DOX (4 mg/kg) and DEXRA (40 mg/kg), or vehicle (10 mL/kg of a 0.9% NaCl solution; B. Braun, Belgium) were intraperitoneally administered to *C57BL/6J* mice. Various protocols of DOX administration have been reported for mice. Injection of a single bolus of 20–25 mg DOX/kg is associated with high mortality and a severe decline in left ventricular ejection fraction (LVEF; decline to a value below 40%) [28, 29]. Alternatively, repeated administration of lower doses (2.5–5 mg DOX/kg for several cycli) result in moderate cardiotoxicity (LVEF decline of 10–15%) [30]. Since we aimed to investigate vascular (dys)function in the absence of cardiotoxicity, we used an *in vivo* dose of 4 mg/kg. We have previously shown that a single DOX dose of 4 mg/kg does not adversely impact LVEF [25]. A DEXRA dose of 40 mg/kg was administered to resemble the clinically recommended DEXRA/DOX ratio of 10 [27]. Of note, DEXRA was administered 30 minutes before DOX, as performed in patients [27]. After 16h, mice were euthanised with sodium pentobarbital (200 mg/kg, intraperitoneal; Sanofi) and the thoracic aorta was isolated for vascular reactivity evaluation.

**Ex vivo.** For *ex vivo* mechanistic experiments, mice were euthanised with sodium pentobarbital (200 mg/kg, intraperitoneal; Sanofi), followed by perforation of the diaphragm (when under deep anaesthesia) to isolate the thoracic aorta. Following isolation, the aorta was dissected into segments of 2 mm length and immediately transferred to DMEM medium (Thermo Fisher Scientific) supplemented with 1% penicillin/streptomycin (Thermo Fisher Scientific) and 1% foetal bovine serum (Thermo Fisher Scientific). Aortic segments were randomly divided and received either PBS as vehicle (1:100), DOX (1 μM), a combination of DOX (1 μM) and DEXRA (10 μM), or a combination of DOX (1 μM) and the antioxidant N-acetylcysteine (NAC; 10 μM) for 16h. A NAC dose of 10 μM was chosen as a NAC to DOX ratio of 10 to 1 has shown effective capability in attenuating DOX-induced ROS formation and prevent cardiotoxicity in both patients and rodent experimental models [31–33]. DEXRA and NAC were administered 15 minutes prior to DOX addition to allow the uptake of DEXRA and NAC first. After 16h, aortic segments were mounted between two hooks of an organ bath set-up (10 mL) filled with Krebs Ringer solution (37˚C, 95% $O_2$/5% $CO_2$, pH 7.4) for vascular reactivity evaluation.

## Evaluation of vascular reactivity

Aortic segments were mounted at a preload of 20 mN, and the experimental protocol was started 20 minutes thereafter to allow optimal stabilisation. VSMC contraction was evaluated by adding a single dose of phenylephrine (PE; 2 μM), an α1-adrenergic receptor agonist, for 15 minutes. Next, endothelium-dependent vasodilation was investigated by addition of cumulative concentrations of acetylcholine (ACh; 3 nM–10 μM), a muscarinic receptor agonist. After washing steps to remove PE and ACh, PE-stimulated contraction was repeated, and, once stable, Nω-nitro-L-arginine methyl ester (L-NAME; 300 μM), an inhibitor of endothelial nitric oxide synthase (eNOS), was added. After 20 minutes, cumulative concentrations of the exogenous nitric oxide (NO)-donor diethylamine NONOate (DEANO; 0.3 nM–10 μM) were added to the organ bath to evaluate endothelium-independent vasodilation of VSMCs through the cGMP-mediated pathway.

## RNA isolation, DNA reverse transcription and RT-qPCR

RNA was isolated and purified from two thoracic aortic segments (each 2 mm in length) and cardiac tissue using the RNeasy® Micro Kit (QIAGEN) and ISOLATE II RNA Mini Kit (Meridian Bioscience®), respectively, according to the manufacturer's instructions. RNA

purity and concentration were measured with the NanoDrop® ND-1000 spectrophotometer. Next, reverse transcription was performed using the TaqMan™ Reverse Transcription Reagents kit (Thermo Fisher Scientific; Invitrogen), according to the manufacturer's instructions. Random hexamers (2.5 μM), provided with this kit, were used as primers. Finally, RT-qPCR was performed with the TaqMan™ Fast Advanced Master Mix (Thermo Fisher Scientific) and the primers Mm00493776_m1 for *Top-IIβ*, Mm00607939_s1 for *β-actin* and Mm99999915_g1 for *Gapdh* (all from Thermo Fisher Scientific) in a QuantStudio™ 3 Real-Time PCR system (Applied Biosystems™). The run method consisted of an activation step (2 minutes at 50˚C followed by 10 minutes at 95˚C) and 45 cycles of 15 seconds at 95˚C and 1 minute at 60˚C.

## Western blotting

Cardiac and thoracic aorta samples obtained from the *in vivo* experiments were used for all Western blotting (WB) experiments. Cardiac tissue was homogenised in RIPA lysis buffer (abcam), supplemented with 1 tablet of protease inhibitor (cOmplete Mini, Roche) and 1 tablet of phosphatase inhibitor PhosSTOP (EASYpack, Roche), using the Precellys® 24. Following protein concentration determination with a BCA assay (Thermo Fisher Scientific), samples were diluted in Laemmli buffer (Bio-Rad) containing 5% β-mercaptoethanol (Sigma-Aldrich) to reach a final concentration of 1 μg/μL. For thoracic aorta samples, one 2 mm segment for each condition was directly lysed in Laemmli sample buffer containing 5% β-mercaptoethanol. All samples were subsequently heat-denatured for 5 min at 100˚C. Next, samples were loaded on Bolt 4–12% Bis-Tris gels (Invitrogen) and after electrophoresis transferred to Immobilon-FL PVDF membranes (Merck). After blocking (1h, Odyssey Li−COR blocking buffer (Li-COR Biosciences)), membranes were probed with primary antibodies, diluted in Odyssey Li−COR blocking buffer, overnight at 4˚C. The following primary antibodies were used: rabbit anti-TOP-IIβ (ab125297; abcam), mouse anti-endothelial nitric oxide synthase (eNOS; ab76198; abcam), rabbit anti-phosphorylated eNOS on position serine 1177 (Ser1177-eNOS; ab215717; abcam) and rabbit anti-GAPDH (14C10; Cell Signalling). The next day, membranes were incubated with IRDye-labeled secondary goat anti-rabbit IgG926-68171 (Li−COR Biosciences) and goat anti-mouse IgG926-32210 (Li−COR Biosciences) for 1h at room temperature. Membranes were visualised with an Odyssey SA infrared imaging system (Li−COR Biosciences).

## Chemical compounds

DOX (Adriamycin®, 2 mg/mL) was purchased from Pfizer (Puurs, Belgium). PE, L-NAME, ACh and DEANO were obtained from Sigma-Aldrich (Overijse, Belgium). DEXRA and NAC were purchased from Tocris (Bio-Techne, Dublin, Ireland).

## Statistical analysis

All results are expressed as the mean ± standard error of the mean (SEM). Statistical analyses were performed using GraphPad Software (Prism 10—Version 10.0.1; Graphpad, California, United States of America). A p-value < 0.05 was considered statistically significant. The following statistical tests were performed: For cell viability experiments, a one-way ANOVA with Tukey's multiple comparisons test; for ACh and DEANO curves, a repeated measures two-way ANOVA with Dunnett's multiple comparisons test; for PE contraction with and without L-NAME, a one-way ANOVA with Dunnett's multiple comparisons test; and for RT-qPCR and WB, a Mann-Whitney U test.

## Results

First, we validated DEXRA efficacy in DOX-treated cardiomyocytes (n = 3 biological replicates). After 16h, DOX reduced cardiomyocyte viability (69.0 ± 3.6% in DOX groups vs. 95.7 ± 1.2% in vehicle and 96.3 ± 0.9% in DEXRA groups) (S1A Fig). DOX-induced cardiomyocyte death was mitigated by DEXRA (87.3 ± 3.1%), implying that DEXRA is active. Cardiomyocyte viability between the vehicle and DEXRA groups was similar (S1A Fig). Furthermore, in ECs and VSMCs (n = 3 biological replicates for both), cell viability did not differ between all treatment groups (S1B and S1C Fig).

Next, vascular reactivity was evaluated at 16h after DOX administration (4 mg/kg) to mice. ACh-stimulated endothelium-dependent vasodilation was impaired in the DOX-treated group (Fig 1A). Pre-treatment with DEXRA (40 mg/kg) did not prevent DOX-induced endothelial dysfunction (Fig 1A). DEANO-stimulated endothelium-independent vasodilation was similar between all groups (Fig 1A). Finally, PE-induced contraction did not differ between the treatment groups in the absence and presence of L-NAME (Fig 1B).

To further corroborate the *in vivo* findings, experiments were repeated *ex vivo*. Similar to the *in vivo* observations, DOX (1 μM) impeded ACh-stimulated vasodilation after 16h, regardless of DEXRA (10 μM) pre-incubation (Fig 2A). Moreover, DEANO-induced vasodilation curves were identical between all treatment groups (Fig 2A). In contrast to the *in vivo* setting, *ex vivo* DOX treatment resulted in lower PE-induced contraction, irrespective of L-NAME addition (Fig 2B).

We subsequently investigated the role of ROS and TOP-IIβ in DOX-induced vascular toxicity. Regardless of pre-incubation of aortic segments *ex vivo* with the ROS scavenger NAC (10 μM), DOX impaired ACh-induced vasodilation (Fig 3A). Vasodilation curves with DEANO did not differ between the treatment groups (Fig 3A). NAC did not prevent the

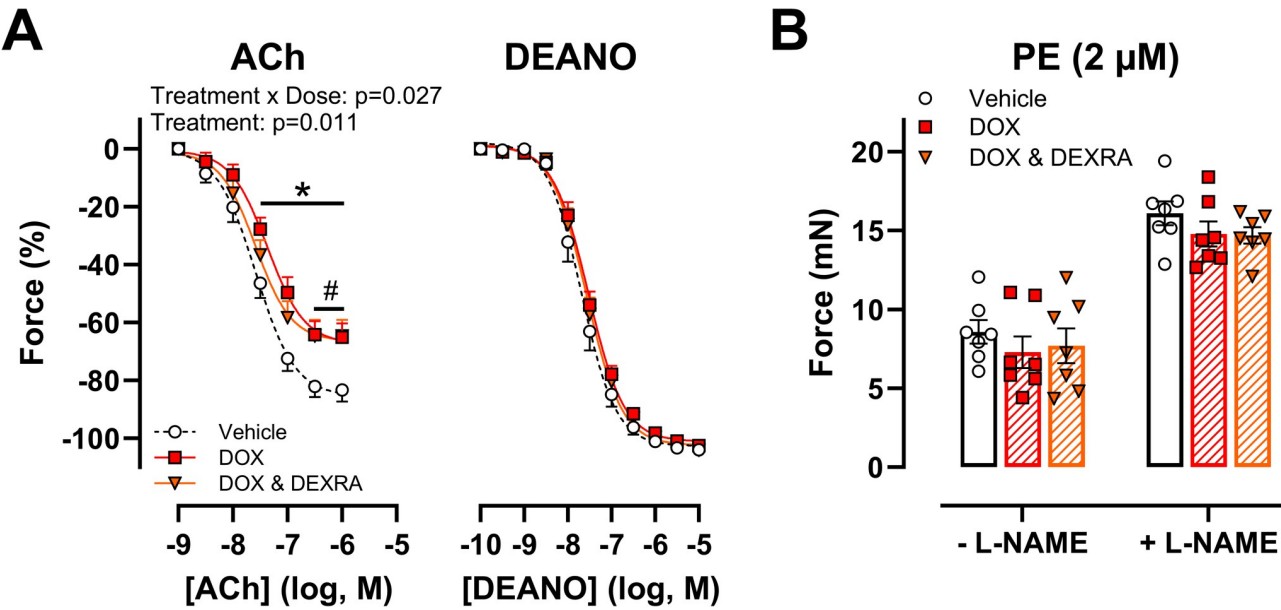

**Fig 1. Evaluation of vascular reactivity, following short-term in vivo DOX and DEXRA treatment.** DOX impaired ACh-induced vasodilation, irrespective of DEXRA pre-treatment (A). DEANO-induced vasodilation did not differ between the treatment groups (A). In both the absence and presence of L-NAME, PE-induced contraction remained unaffected (B). For A: Repeated measures two-way ANOVA with Dunnett's multiple comparisons test. For B: One-way ANOVA with Dunnett's multiple comparisons test per L-NAME condition. n = 7 in each group. For A: *p<0.05 for vehicle vs. DOX group; #p<0.05 for vehicle vs. DOX with DEXRA group. For B: p>0.05 for vehicle vs. DOX group and vehicle vs. DOX with DEXRA group.

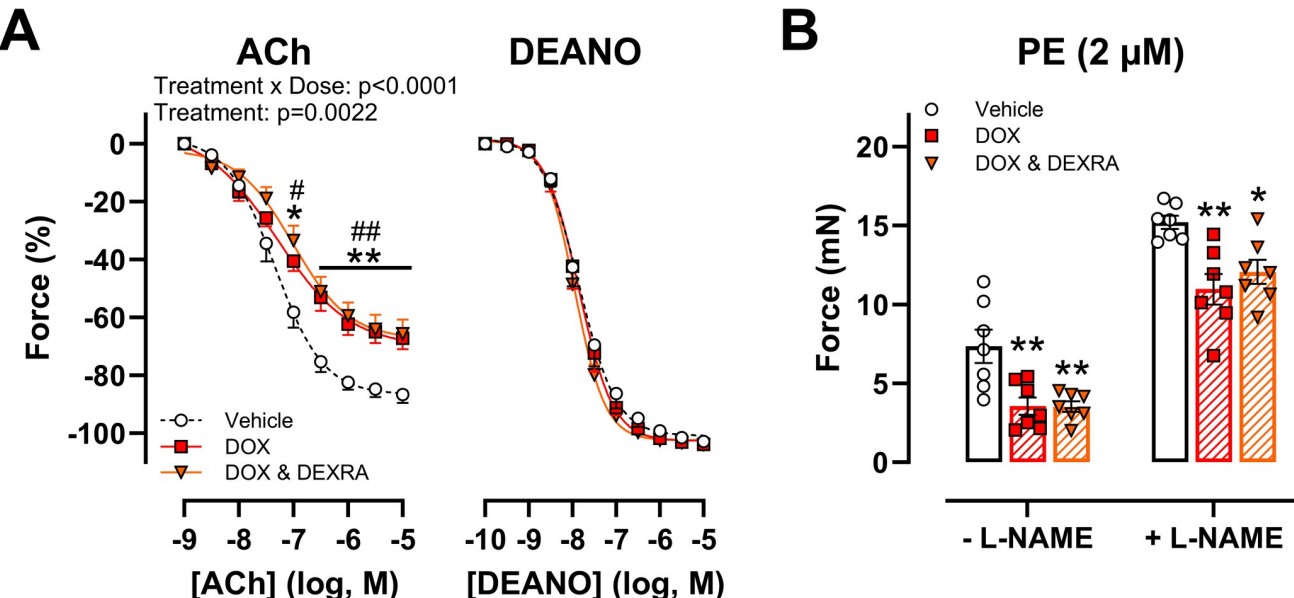

**Fig 2. Evaluation of vascular reactivity, following short-term ex vivo DOX and DEXRA incubation.** ACh-induced vasodilation was diminished in the DOX and DOX with DEXRA groups (A). DEANO-induced vasodilation was similar between the treatment groups (A). PE-induced contraction with and without L-NAME was lower following DOX exposure (B). For A: Repeated measures two-way ANOVA with Dunnett's multiple comparisons test. For B: One-way ANOVA with Dunnett's multiple comparisons test per L-NAME condition. n = 7 in each group. For A: *, **p<0.05, 0.01 for vehicle vs. DOX group; #, ##p<0.05, 0.01 for vehicle vs. DOX with DEXRA group. For B: *,**p<0.05, 0.01 compared to vehicle.

decrease in PE-induced contraction in the DOX and DOX with DEXRA groups either (Fig 3B). With regards to TOP-IIβ, DOX did not alter TOP-IIβ expression in the heart and thoracic aorta, following 16h of *in vivo* treatment (Fig 4A). On the other hand, TOP-IIβ protein levels were lower in the DOX-treated group in cardiac, but not aortic, tissue (Fig 4B). Of note, cardiac tissue, which is known to express TOP-IIβ, was used as a positive control. Full, unedited blots are shown in the S1 Raw images.

Finally, we examined whether DOX-induced endothelial dysfunction is attributable to altered eNOS expression in the thoracic aorta. Both eNOS and Ser1177-eNOS levels remained unaffected, 16h after *in vivo* DOX administration (S2A Fig). Full, unedited blots are shown in the S1 Raw images.

## Discussion

The present study is the first to evaluate the therapeutic potential of DEXRA against DOX-related vascular toxicity in a previously-established *in vivo* and *ex vivo* model of vascular dysfunction induced by 16h DOX treatment [25]. *In vivo* and *ex vivo* treatment with DOX provoked endothelial dysfunction and VSMC hypocontraction, as evidenced by impaired ACh-stimulated vasodilation and diminished PE-induced contraction, respectively. These results are in line with our previous observations [25], thus highlighting the reproducibility of the *in vivo* and *ex vivo* models. Strikingly, in this experimental model, DEXRA failed to prevent both endothelial dysfunction and VSMC hypocontraction provoked by DOX. This suggests that, in contrast to cardiotoxicity, DEXRA does not protect against DOX-induced vascular toxicity, at least in the evaluated acute setting. However, as we have previously shown that DOX-induced vascular toxicity displays a dynamic profile, characterised by initial hypocontraction (after 16h) [25], subsequent hypercontraction (after 6 days) [11] and an eventual increase in pro-inflammatory (glyco)proteins (after 6 injections) [34], further research investigating the

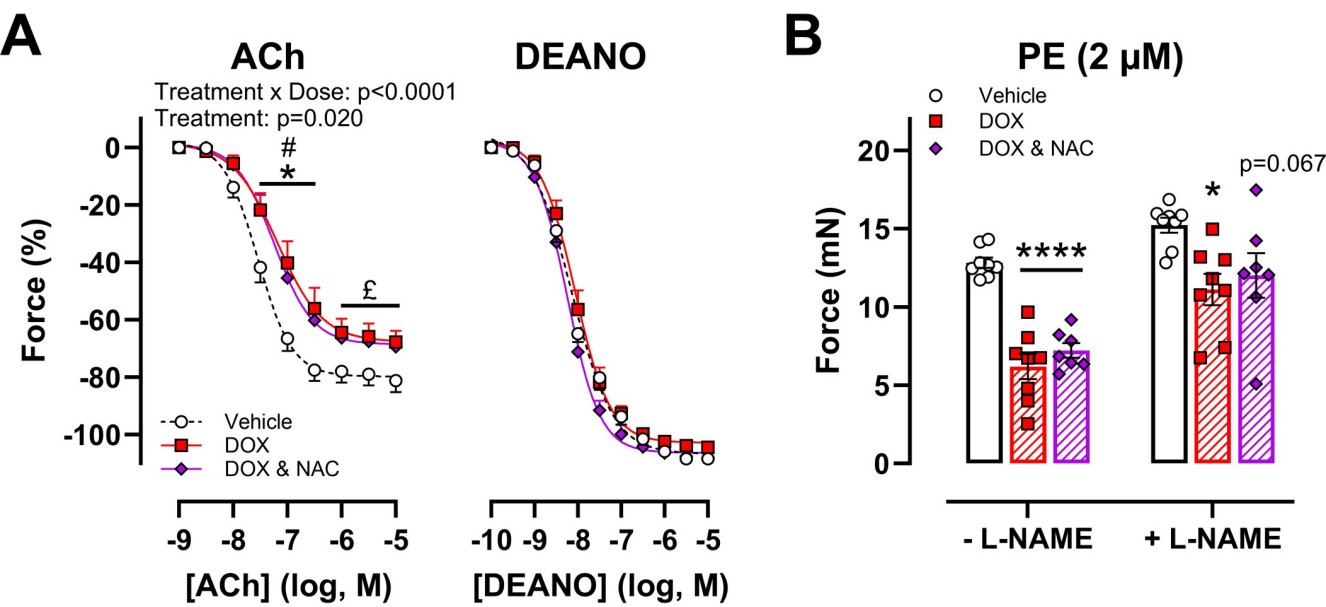

**Fig 3. Evaluation of vascular reactivity, following short-term ex vivo DOX and NAC incubation.** DOX diminished ACh-induced vasodilation, irrespective of NAC (A). DEANO-induced vasodilation did not differ between the treatment groups (A). In both the absence and presence of L-NAME, PE-induced contraction was lower in the DOX group, regardless of NAC (B). For A: Repeated measures two-way ANOVA with Dunnett's multiple comparisons test. For B: One-way ANOVA with Dunnett's multiple comparisons test per L-NAME condition. n = 8 in vehicle and DOX groups; n = 7 in DOX with NAC group. For A: *p<0.05 for vehicle vs. DOX group; #p<0.05 for vehicle vs. DOX with NAC group; £ 0.05<p<0.08 for vehicle vs. DOX groups and for vehicle vs. DOX with NAC groups. For B: *, ****p<0.05, 0.0001 compared to vehicle.

protective capability of DEXRA against vascular toxicity from longer-term repetitive DOX dosing is still warranted.

It is surprising that DEXRA did not prevent vascular dysfunction following DOX exposure as it is tempting to assume that DOX provokes vascular toxicity in a similar way as cardiotoxicity. The cardioprotective action of DEXRA is typically ascribed to its depletion of TOP-IIβ [20, 21]. While our data show a similar decline in TOP-IIβ protein levels in the heart of DOX-treated mice, TOP-IIβ protein levels conversely did not change in aortic tissue. Alternatively, the antioxidative capacity of DEXRA has been proposed to contribute to its cardioprotective action as well [24], although its antioxidant profile has been challenged in recent years [20, 21]. To further assess the possible contribution of ROS to DOX-induced vascular toxicity, *ex vivo* experiments were repeated with NAC. NAC (pre-)treatment did not prevent endothelial dysfunction nor VSMC hypocontraction. Collectively, these data suggest that, at least in the evaluated model, DOX induces vascular toxicity independently from TOP-IIβ and ROS-mediated pathways, which may explain why DEXRA pre-treatment had no beneficial effect.

To further understand the mechanisms involved in DOX-induced endothelial dysfunction, eNOS levels and phosphorylation of eNOS on its active site Ser1177 (i.e. Ser1177-eNOS) were evaluated. DOX did not alter eNOS protein levels nor Ser1177-eNOS levels. This differs from our previous work reporting decreased eNOS expression (after 6 days) [11], and from a study by He et al. who observed reduced eNOS and Ser1177-eNOS levels after 3 weeks of DOX (2.5 mg/kg over 3 weeks; 15 mg/kg cumulative) [12], again raising awareness for the influence of timing and repetitive DOX administration on vascular toxicity. Although it remains unclear how DOX provokes endothelial dysfunction in the current work, we previously observed that 16h DOX treatment disturbs $Ca^{2+}$ influx over non-selective cation channels in VSMCs,

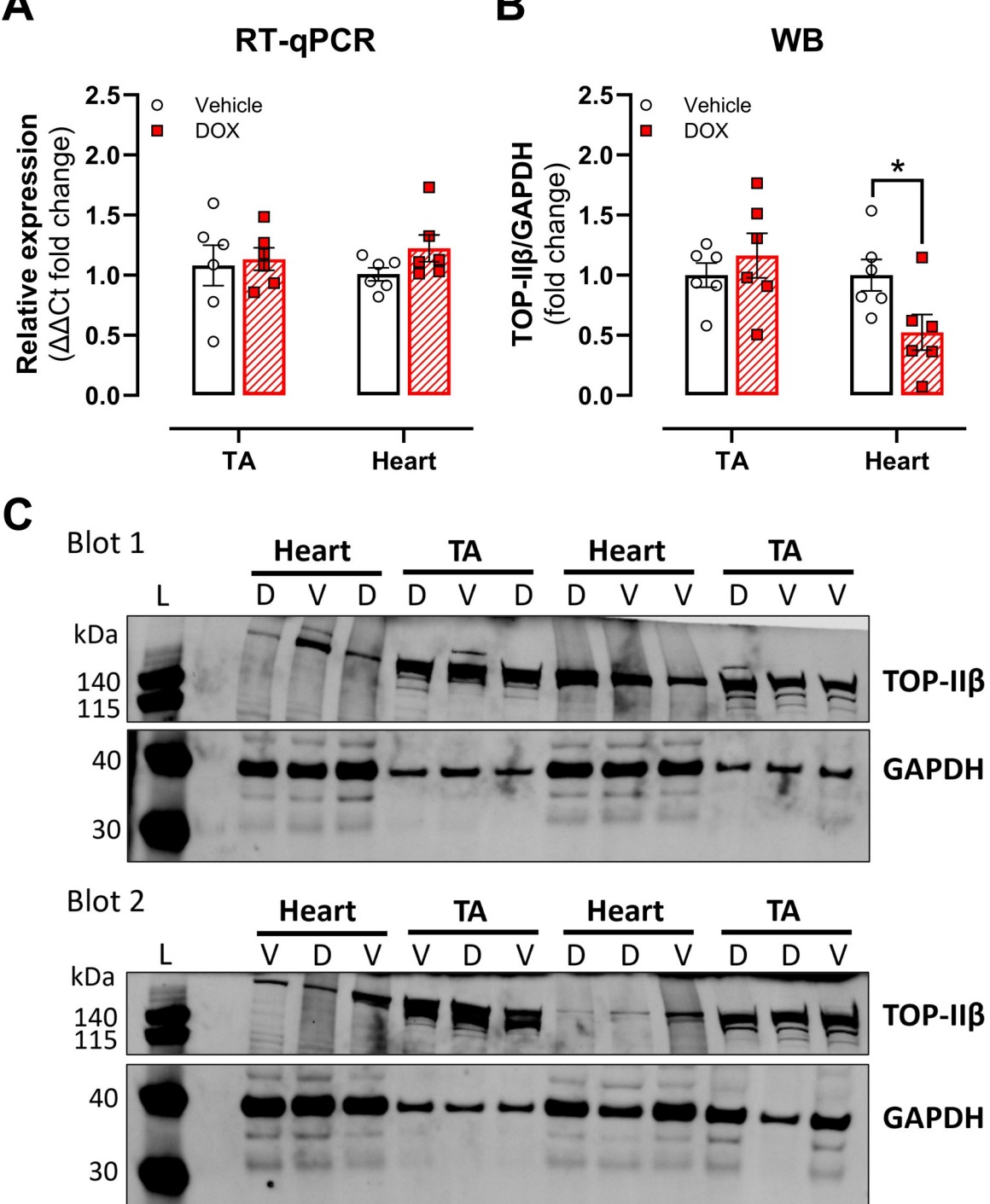

**Fig 4. Assessment of TOP-IIβ expression in the murine thoracic aorta (TA) and heart, following short-term in vivo DOX treatment.** RT-qPCR (A) and WB (B) show that murine aortic tissue expresses TOP-IIβ. DOX does not alter TOP-IIβ expression in the heart and aorta (A). DOX decreased TOP-IIβ protein levels in cardiac, but not aortic, tissue (B). Representative blots for panel B (C). For A & B: Mann-Whitney U test. n = 6 in each group. p>0.05 for vehicle vs. DOX groups in panel A. *p<0.05 for vehicle vs. DOX groups in panel B. For C: "L", "V" and "D" stand for ladder, vehicle and DOX, respectively.

thereby attenuating contraction [25]. Since activation of eNOS depends on $Ca^{2+}$ influx [35, 36], it is possible that DOX similarly impairs endothelial function after 16h.

In summary, shortly after DOX administration, DEXRA failed to mitigate associated vascular toxicity, both *in vivo* and *ex vivo*. Furthermore, our data suggest that such acute DOX treatment provokes vascular toxicity independently from TOP-IIβ and ROS, which may explain the inaptitude of DEXRA to mitigate associated vascular dysfunction. However, further study is warranted that evaluates the protective capability of DEXRA against vascular toxicity caused by longer-term repetitive DOX administration.

## Limitations

The present study has some limitations. First, the present study focusses on the acute adverse effects of DOX on the vasculature (i.e. 16h after administration), which may differ from longer-term effects of repetitive dosing. Second, our findings only apply to young, male mice, which may differ from female and old mice. Young, male mice were chosen as they are more sensitive to DOX-induced cardiovascular toxicity [37–39], and to avoid the influence of cyclic changes in female hormones as confounding factors. Finally, the intraperitoneal administration route in the current work differs from the clinical setting where DOX is administered intravenously.

## Supporting information

**S1 Fig. Viability of cardiomyocytes, endothelial cells (ECs) and vascular smooth muscle cells (VSMCs), following short-term DOX and DEXRA incubation.** DOX-treated cardiomyocytes showed lower viability compared to the vehicle- and DEXRA-treated groups, but this was attenuated in the presence of DEXRA (A). DOX did not affect viability in ECs (B) and VSMCs (C). For all panels: One-way ANOVA with Tukey's multiple comparisons test; n = 3 in each group. For A: *, ****$p < 0.05$, 0.0001 compared to vehicle; #, #### $p < 0.05$, 0.0001 compared to DEXRA group; £££ $p < 0.001$ between DOX and DOX with DEXRA groups. For B & C: $p > 0.05$ for all groups.
(TIF)

**S2 Fig. Assessment of eNOS and Ser1177-eNOS levels in the murine thoracic aorta, following short-term in vivo DOX treatment.** DOX does not alter eNOS nor Ser1177-eNOS levels shortly after administration (A). Representative blot for panel A (B). For A: Mann-Whitney U test. n = 6 in each group. $p > 0.05$ for vehicle vs. DOX groups. For B: "L", "V" and "D" stand for ladder, vehicle and DOX, respectively.
(TIF)

**S1 File.**
(ZIP)

**S1 Raw images.**
(PDF)

## Author Contributions

**Conceptualization:** Matthias Bosman, Emeline M. Van Craenenbroeck, Pieter-Jan Guns.

**Formal analysis:** Matthias Bosman, Dustin N. Krüger, Pieter-Jan Guns.

**Funding acquisition:** Guido R. Y. De Meyer, Emeline M. Van Craenenbroeck, Pieter-Jan Guns.

**Investigation:** Matthias Bosman, Dustin N. Krüger.

**Methodology:** Matthias Bosman.

**Supervision:** Emeline M. Van Craenenbroeck, Pieter-Jan Guns.

**Visualization:** Matthias Bosman.

**Writing – original draft:** Matthias Bosman, Emeline M. Van Craenenbroeck, Pieter-Jan Guns.

**Writing – review & editing:** Matthias Bosman, Dustin N. Krüger, Kasper Favere, Guido R. Y. De Meyer, Constantijn Franssen, Emeline M. Van Craenenbroeck, Pieter-Jan Guns.

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
