## [Decision Letter · Decision Letter 0]

1 Jun 2023

PONE-D-23-14227Dexrazoxane does not mitigate doxorubicin-induced endothelial dysfunction in an ex vivo model of acute vascular toxicityPLOS ONE

Dear Dr. Bosman,

Thank you for submitting your manuscript to PLOS ONE. After careful consideration, we feel that it has merit but does not fully meet PLOS ONE’s publication criteria as it currently stands. Therefore, we invite you to submit a revised version of the manuscript that addresses the points raised during the review process.

We look forward to receiving your revised manuscript.

Kind regards,

Peng Zhang, Ph.D.

Academic Editor

PLOS ONE

Journal Requirements:

Additional Editor Comments:

Please carefully reponse the reviewer's comments.

Reviewers' comments:

Reviewer's Responses to Questions

**Comments to the Author**

1. Is the manuscript technically sound, and do the data support the conclusions?

Reviewer #1: Partly

Reviewer #2: Yes

2. Has the statistical analysis been performed appropriately and rigorously? 

Reviewer #1: Yes

Reviewer #2: I Don't Know

3. Have the authors made all data underlying the findings in their manuscript fully available?

Reviewer #1: No

Reviewer #2: No

4. Is the manuscript presented in an intelligible fashion and written in standard English?

Reviewer #1: Yes

Reviewer #2: Yes

5. Review Comments to the Author

Reviewer #1: The manuscript titled, “Dexrazoxane does not mitigate doxorubicin-induced endothelial dysfunction in an ex vivo model of acute vascular toxicity” describes a study in which investigators sought to determine the influence of the only FDA-approved drug for mitigating doxorubicin (doxo)-induced cardiotoxicity on vascular dysfunction with doxo. Vascular dysfunction is an independent risk factor for future overt cardiovascular/cardiac events; thus, the biomedical rationale for this study is sound. The major strength of this study is data regarding the repurposing of an existing drug with known therapeutic effects. There are some limitations of the study that limit enthusiasm for this manuscript in its current form. The comments provided below are intended to improve the overall scientific merit of the manuscript.

- Mice were already being sacrificed for these studies, so why not administer DOXO and DEXRA in vivo and then excise the arteries to look at function? This is more indicative of the clinical condition. Moreover, if authors detected differences (or lack thereof) in vivo, it then would make sense to go to the ex vivo model

- Provide rationale for DOX and DEXRA concentrations used in culture experiments. Authors provide some rationale for the timing, but the rationale is lacking for concentration. Although authors do provide rationale for the timing, did authors account for timing differences in vivo vs in vitro/ex vivo – e.g., the 15min rationale is clear for in vivo studies, but might it need to be shorter ex vivo? Also, why 16 hours in the in vitro & ex vivo experiments?

o Rationale for timing and concentrations are provided in the discussion, but it would be helpful from a rigor and reproducibility perspective to include that information in the methods

- Please provide a clear description of the animal chow.

- Why were only male mice used?

- Topoisomerase II has shown to be expressed in arteries – why do authors think they were unable to detect it? Given that authors incubated the Topo-II antibody along with beta actin (vs. separately) it could be that beta actin was expressed lower in cardiomyocytes vs. aorta. In support of this point, beta actin abundance is far lower in the cardiomyocytes than in aortas, and given the co-incubation of antibodies, it could be that the beta actin antibody concentration should be adjusted across sample types. Before stating that topoisomerase II is not expressed in the vasculature, authors should consider these experimental considerations.

- Cell viability assays are described in the methods. Authors should show results for these assays

- Vessel assays with L-NAME are described in the methods. Authors should show results for these assays

- Authors do not present data to support this statement and the interpretation of these results are not completely accurate. Lines 180-183….“However, our data do not support that DOX impairs endothelial function through ROS-mediated depletion of NO since DEANO-induced vasodilation curves were aligned in all treatment 183 groups, implying that exogeneous NO is not captured and converted into peroxynitrite.” DEANO is an exogenous NO donor and results of those studies does not provide insight into the role of ROS-related suppression of NO bioavailability. To support this statement, authors would need to show that DOX-mediated endothelial function is not improved with co-incubation of DOXO with a ROS scavenger – these data would show a cause-and-effect role of ROS in mediating endothelial dysfunction with DOXO. This manuscript (PMID: 33073250) demonstrates a role for ROS in mediating endothelial dysfunction with DOXO, so authors should consider the findings from this manuscript when interpreting their results.

Reviewer #2: Doxorubicin (DOX) chemotherapy is well-known to detrimentally affect the cardiovascular system. The only approved drug is (DEXRA), a TOP-IIbeta inhibitor which can prevent cardiac toxicity, but it is unknown if it can also protect the vasculature. The authors showed that DEXRA protects against acute DOX toxicity (cell death) in cultured cardiomyocytes, but that it does not protect against acutely impaired vascular endothelial function in isolated aortas. This discrepancy is possibly due to the lack of TOP-IIbeta expression in aortas vs. the heart. Overall, the manuscript is straightforward, and the conclusions are supported. However, there are some limitations and concerns, as follows:

• Although the ex vivo approaches are innovative, they limit the investigation to acute administration of DOX and DEXRA, which is a limitation as the mechanisms of action may be different (the authors have previously shown that eNOS expression is reduced with 2 weeks of in vivo DOXO but it was not in the present study with acute administration). That said, if DEXRA only works through inhibiting TOP-IIbeta, acute vs. chronic doesn’t matter.

• Why weren’t the cell viability experiments also performed in vascular cells? This seems like something that would have been easy to do and would allow for more direct comparisons between effects of DEXRA in the heart vs. vasculature. Also note, that it w

• What is the sample size for Top-11beta protein expression? The authors should include a quantification that includes the full sample. Does DOX increase TOP-IIbeta expression? If not, how do you know that the beneficial effect of DEXRA in your experimental setup (in cardiomyocytes) is mediated by depleting TOP-IIbeta? It would also have been helpful to include a DOX + DEXRA condition to know if DEXRA is inhibiting TOP-IIbeta within this period and the ex vivo experimental setup.

• The manuscript seems very under referenced, with a high percentage of the only 11 citations coming from the authors. There are a large number of papers now investigated cardio-toxic effects of DOX and potential interventions to mitigate it that have been omitted.

• Lines 192-193: This should be tempered. You cannot conclude from your data that DOX impaired endothelial function by influencing eNOS. E.g., DOX could be lowering NO-dependent dilation via greater ROS scavenging.

• Statistical tests used should be described in the “Methods – Statistical Analysis” section.

6. PLOS authors have the option to publish the peer review history of their article (what does this mean?). If published, this will include your full peer review and any attached files.

Reviewer #1: No

Reviewer #2: No

---

## [Author Response · Author response to Decision Letter 0]

13 Oct 2023

Rebuttal for manuscript PONE-D-23-14227-R1 

We would like to thank the editor and reviewers for the valuable comments and for the opportunity to resubmit our manuscript. To address the feedback of the reviewers, additional experiments and analyses have been performed. 

In summary:

1. Throughout the manuscript (including the title), we have acknowledged that our results and conclusions relate to (very) early DOX-induced vascular toxicity (16h), and that we cannot exclude a protective role for DEXRA in longer-term vascular toxicity.

2. As requested by the reviewers, an in vivo study was performed where DOX (4 mg/kg) and DEXRA (40 mg/kg) were administered intraperitoneally to C57BL6/J mice, and vascular reactivity was evaluated 16 hours (h) later.

3. TOP-IIβ expression levels were investigated in the in vivo samples (vehicle and DOX) using both qPCR and Western blotting.

4. To delineate the role of ROS in DOX-induced endothelial dysfunction, the antioxidant N-acetylcysteine (NAC) was used in the ex vivo model where isolated murine aortic segments were treated with either vehicle, DOX (1 μM) or a combination of DOX (1 μM) with NAC (10 μM) for 16h. 

5. Finally, the effect of DOX on eNOS expression in the in vivo samples was investigated with Western blotting.

As a result, substantial parts of the manuscript have been reorganised and rewritten, and new figures were created. Moreover, the manuscript’s title has been changed and a Limitations part was added to the Discussion section of the manuscript. For the convenience of the reviewers, we have included two versions of the revised manuscript, namely PONE-D-23-14277-R1 that contains the changes highlighted in yellow and PONE-D-23-14277-R1 _CLEAN that represents the new manuscript only. 

Please find point-by-point responses to the reviewer comments in the rebuttal below. Line numbers refer to PONE-D-23-14277-R1 _CLEAN.

Reviewer #1

1. Mice were already being sacrificed for these studies, so why not administer DOXO and DEXRA in vivo and then excise the arteries to look at function? This is more indicative of the clinical condition. Moreover, if authors detected differences (or lack thereof) in vivo, it then would make sense to go to the ex vivo model

The reviewer makes a valid point. We have added an in vivo study where DOX (4 mg/kg) and DEXRA (40 mg/kg) were administered intraperitoneally to C57BL6/J mice. The in vivo results were in line with the ex vivo observations. Specifically, DEXRA failed to restore endothelial dysfunction induced by short-term in vivo DOX exposure.

We have discussed the in vivo experiment in the Results section as follows (from line 193 to 198, p. 9):

“Next, vascular reactivity was evaluated at 16h after DOX administration (4 mg/kg) to mice. ACh-stimulated endothelium-dependent vasodilation was impaired in the DOX-treated group (Fig 1A). Pre-treatment with DEXRA (40 mg/kg) did not prevent DOX-induced endothelial dysfunction (Fig 1A). DEANO-stimulated endothelium-independent vasodilation was similar between all groups (Fig 1A). Finally, PE-induced contraction did not differ between the treatment groups in the absence and presence of L-NAME (Fig 1B).”

Fig 1: Evaluation of vascular reactivity, following short-term in vivo DOX and DEXRA treatment. DOX impaired ACh-induced vasodilation, irrespective of DEXRA pre-treatment (A). DEANO-induced vasodilation did not differ between the treatment groups (A). In both the absence and presence of L-NAME, PE-induced contraction remained unaffected (B). For A: Repeated measures two-way ANOVA with Dunnett’s multiple comparisons test. For B: One-way ANOVA with Dunnett’s multiple comparisons test per L-NAME condition. n=7 in each group. For A: *p<0.05 for vehicle vs. DOX group; #p<0.05 for vehicle vs. DOX with DEXRA group. For B: p>0.05 for vehicle vs. DOX group and vehicle vs. DOX with DEXRA group.

2. Provide rationale for DOX and DEXRA concentrations used in culture experiments. Authors provide some rationale for the timing, but the rationale is lacking for concentration. Although authors do provide rationale for the timing, did authors account for timing differences in vivo vs in vitro/ex vivo – e.g., the 15min rationale is clear for in vivo studies, but might it need to be shorter ex vivo? Also, why 16 hours in the in vitro & ex vivo experiments? Rationale for timing and concentrations are provided in the discussion, but it would be helpful from a rigor and reproducibility perspective to include that information in the methods

For the in vivo and in vitro/ex vivo experiments, we have explained the choice of the DOX and DEXRA doses of 4 and 40 mg/kg, respectively, in the Materials & Methods section as follows (from line 96 to 105, p. 5): 

“Various protocols of DOX administration have been reported for mice. Injection of a single bolus of 20-25 mg DOX/kg is associated with high mortality and a severe decline in left ventricular ejection fraction (LVEF; decline to a value below 40%) [1, 2]. Alternatively, repeated administration of lower doses (2.5-5 mg DOX/kg for several cycli) result in moderate cardiotoxicity (LVEF decline of 10-15%) [3]. Since we aimed to investigate vascular (dys)function in the absence of cardiotoxicity, we used an in vivo dose of 4 mg/kg. We have previously shown that a single DOX dose of 4 mg/kg does not adversely impact LVEF [4]. A DEXRA dose of 40 mg/kg was administered to resemble the clinically recommended DEXRA/DOX ratio of 10 [5]. Of note, DEXRA was administered 30 minutes before DOX, as performed in patients [5].”,

from line 89 to 90, p. 4: 

“The DOX dose of 1 μM was chosen to resemble plasma concentrations in patients [6]. A DEXRA dose of 10 μM was used since, in patients, a dosage ratio (DEXRA to DOX) of 10 to 1 is recommended [5].”

and from line 116 to 118, p. 5-6: 

“A NAC dose of 10 μM was chosen as a NAC to DOX ratio of 10 to 1 has shown effective capability in attenuating DOX-induced ROS formation and prevent cardiotoxicity in both patients and rodent experimental models [7-9].” 

Furthermore, we would like to note that DEXRA administration to mice 30 minutes prior to DOX, as performed in patients, did not prevent DOX-induced endothelial dysfunction, which is identical to the ex vivo model where DEXRA was administered 15 minutes before DOX. As such, the time frame for DEXRA and DOX addition ex vivo is an appropriate reflection of the in vivo model.

Finally, the reasoning for the 16h treatment period is explained in the Materials & Methods section (from line 85 to 89, p. 4): 

“The same treatment period (16h) was used for the in vivo and ex vivo experiments (see further) as we have previously shown that, at this time point, DOX provokes vascular dysfunction in vivo and ex vivo [4], thus representing a readily-accessible model of acute DOX-induced vascular toxicity.”

3. Please provide a clear description of the animal chow.

This information was added in the Materials & Methods section as follows (from line 73 to 75, p. 4):

“Male C57BL/6J mice (age: 10-12 weeks; body weight: 24-30 g; Charles River) were housed in the animal facility of the University of Antwerp in standard cages with 12–12 h light–dark cycles with access to water ad libitum and regular chow, containing in %: crude protein, 19.0; crude fat, 3.3; crude fibre, 5.0; crude ash, 6.4; starch, 35.9; sugar, 5.4 and nitrogen-free extracts, 54.6.”

4. Why were only male mice used?

Male mice were chosen as they are more prone to cardiovascular toxicity and to avoid the influence of cyclic changes in female hormones as potential confounding factors. This is acknowledged in the Limitations part of the Discussion section as follows (from line 309 to 312, p. 14): 

“Second, our findings only apply to young, male mice, which may differ from female and old mice. Young, male mice were chosen as they are more sensitive to DOX-induced cardiovascular toxicity [10-12], and to avoid the influence of cyclic changes in female hormones as confounding factors.”

5. Topoisomerase II has shown to be expressed in arteries – why do authors think they were unable to detect it? Given that authors incubated the Topo-II antibody along with beta actin (vs. separately) it could be that beta actin was expressed lower in cardiomyocytes vs. aorta. In support of this point, beta actin abundance is far lower in the cardiomyocytes than in aortas, and given the co-incubation of antibodies, it could be that the beta actin antibody concentration should be adjusted across sample types. Before stating that topoisomerase II is not expressed in the vasculature, authors should consider these experimental considerations.

We thank the reviewer for this pertinent remark. We have performed new Western blots with a new TOP-IIβ antibody (catalogue number ab125297; abcam) that is superior compared to the original one, as evidenced by a band at the correct molecular weight of 140 kDa in both aortic and cardiac tissue. This demonstrates that TOP-IIβ is expressed in aortic tissue. qPCR further confirmed that TOP-IIβ is expressed in aortic tissue. Importantly, TOP-IIβ protein levels were lower in the heart, but not in the aorta, of DOX-treated mice. Due to the considerable differences in β-actin between aortic and cardiac tissue, GAPDH was used for normalisation, which shows more consistent signal across tissues.

The results of these experiments are discussed in the Results section as follows (from line 230 to 234, p. 10-11): 

“With regards to TOP-IIβ, DOX did not alter TOP-IIβ expression in the heart and thoracic aorta, following 16h of in vivo treatment (Fig 4A). TOP-IIβ protein levels were lower in the DOX-treated group in cardiac, but not aortic, tissue (Fig 4B). Of note, cardiac tissue, which is known to express TOP-IIβ, was used as a positive control. Full, unedited blots are shown in the supplementary Unedited Western blots PDF file.”

The results of these experiments are shown in Fig 4, which is provided below as well.

Fig 4: Assessment of TOP-IIβ expression in the murine thoracic aorta (TA) and heart, following short-term in vivo DOX treatment. RT-qPCR (A) and WB (B) show that murine aortic tissue expresses TOP-IIβ. DOX does not alter TOP-IIβ expression in the heart and aorta (A). DOX decreased TOP-IIβ protein levels in cardiac, but not aortic, tissue (B). Representative blots for panel B (C). For A & B: Mann-Whitney U test. n=6 in each group. p>0.05 for vehicle vs. DOX groups in panel A. *p<0.05 for vehicle vs. DOX groups in panel B. For C: “L”, “V” and “D” stand for ladder, vehicle and DOX, respectively.

6. Cell viability assays are described in the methods. Authors should show results for these assays.

The results for the cell viability assays, in cardiomyocytes, endothelial cells (ECs) and vascular smooth muscle cells (VSMCs) have been added in the Results section as follows (from line 186 to 192, p. 9): 

“First, we validated DEXRA efficacy in DOX-treated cardiomyocytes (n=3 biological replicates). After 16h, DOX reduced cardiomyocyte viability (69.0 ± 3.6% in DOX groups vs. 95.7 ± 1.2% in vehicle and 96.3 ± 0.9% in DEXRA groups) (Supplementary Fig 1A). DOX-induced cardiomyocyte death was mitigated by DEXRA (87.3 ± 3.1%), implying that DEXRA is active. Cardiomyocyte viability between the vehicle and DEXRA groups was similar. (Supplementary Fig 1A). Furthermore, in ECs and VSMCs (n=3 biological replicates for both), cell viability did not differ between all treatment groups (Supplementary Fig 1B & 1C).”

The results for the cell viability assays are summarised in Supplementary Fig 1 with the following caption (from line 317 to 324, p. 14):

Supplementary Fig 1: Viability of cardiomyocytes, endothelial cells (ECs) and vascular smooth muscle cells (VSMCs), following short-term DOX and DEXRA incubation. DOX-treated cardiomyocytes showed lower viability compared to the vehicle- and DEXRA-treated groups, but this was attenuated in the presence of DEXRA (A). DOX did not affect viability in ECs (B) and VSMCs (C). For all panels: One-way ANOVA with Tukey’s multiple comparisons test; n=3 in each group. For A: *, ****p<0.05, 0.0001 compared to vehicle; #, #### p<0.05, 0.0001 compared to DEXRA group; £££ p<0.001 between DOX and DOX with DEXRA groups. For B & C: p>0.05 for all groups. 

7. Vessel assays with L-NAME are described in the methods. Authors should show results for these assays

These results have been added.

8. Authors do not present data to support this statement and the interpretation of these results are not completely accurate. Lines 180-183….“However, our data do not support that DOX impairs endothelial function through ROS-mediated depletion of NO since DEANO-induced vasodilation curves were aligned in all treatment 183 groups, implying that exogeneous NO is not captured and converted into peroxynitrite.” DEANO is an exogenous NO donor and results of those studies does not provide insight into the role of ROS-related suppression of NO bioavailability. To support this statement, authors would need to show that DOX-mediated endothelial function is not improved with co-incubation of DOXO with a ROS scavenger – these data would show a cause-and-effect role of ROS in mediating endothelial dysfunction with DOXO. This manuscript (PMID: 33073250) demonstrates a role for ROS in mediating endothelial dysfunction with DOXO, so authors should consider the findings from this manuscript when interpreting their results.

We thank the reviewer for the valuable suggestion. To further evaluate the role of ROS in DOX-induced endothelial dysfunction, the antioxidant N-acetylcysteine (NAC) was used in the ex vivo model. Specifically, aortic segments were treated with either vehicle, DOX or a combination of DOX (1 μM) and NAC (10 μM) for 16h. NAC did not prevent DOX-induced impairment of ACh-stimulated vasodilation, suggesting that DOX impairs endothelial function independently from ROS-mediated pathways, at least in the currently evaluated acute setting.

These results are discussed in the Results section as follows (from line 225 to 229, p. 10): 

“We subsequently investigated the role of ROS and TOP-IIβ in DOX-induced vascular toxicity. Regardless of pre-incubation of aortic segments ex vivo with the ROS scavenger NAC (10 μM), DOX impaired ACh-induced vasodilation (Fig 3A). Vasodilation curves with DEANO did not differ between the treatment groups (Fig 3A). NAC did not prevent the decrease in PE-induced contraction in the DOX and DOX with DEXRA groups either (Fig 3B).”

These results are summarised in Fig 3, which is also shown below.

Fig 3: Evaluation of vascular reactivity, following short-term ex vivo DOX and NAC incubation. DOX diminished ACh-induced vasodilation, irrespective of NAC (A). DEANO-induced vasodilation did not differ between the treatment groups (A). In both the absence and presence of L-NAME, PE-induced contraction was lower in the DOX group, regardless of NAC (B). For A: Repeated measures two-way ANOVA with Dunnett’s multiple comparisons test per L-NAME condition. For B: One-way ANOVA with Dunnett’s multiple comparisons test. n=8 in vehicle and DOX groups; n=7 in DOX with NAC group. For A: *p<0.05 for vehicle vs. DOX group; #p<0.05 for vehicle vs. DOX with NAC group; £ 0.05<p<0.08 for vehicle vs. DOX groups and for vehicle vs. DOX with NAC groups. For B: *, ****p<0.05, 0.0001 compared to vehicle.

Reviewer #2 

• Although the ex vivo approaches are innovative, they limit the investigation to acute administration of DOX and DEXRA, which is a limitation as the mechanisms of action may be different (the authors have previously shown that eNOS expression is reduced with 2 weeks of in vivo DOXO but it was not in the present study with acute administration). That said, if DEXRA only works through inhibiting TOP-IIbeta, acute vs. chronic doesn’t matter.

We agree that there may be a different response between acute and chronic DOX treatment. We have acknowledged this throughout the manuscript, especially in the Discussion as follows (from line 265 to 273, p. 12):

“Strikingly, in this acute setting, DEXRA failed to prevent both endothelial dysfunction and VSMC hypocontraction provoked by DOX. This suggests that, in contrast to cardiotoxicity, DEXRA does not protect against DOX-induced vascular toxicity, at least in the currently evaluated acute setting. However, as we have previously shown that DOX-induced vascular toxicity displays a dynamic profile, characterised by initial hypocontraction (after 16h) [4], subsequent hypercontraction (after 6 days) [13] and an eventual increase in pro-inflammatory (glyco)proteins (after 6 injections) [14], further research investigating the protective capability of DEXRA against vascular toxicity resulting from longer-term repetitive DOX dosing is still warranted.”

And from line 287 to 293, p. 13:

“To further understand the mechanisms involved in DOX-induced endothelial dysfunction, eNOS levels and phosphorylation of eNOS on its active site Ser1177 (i.e. Ser1177-eNOS) were evaluated. DOX did not alter eNOS protein levels nor Ser1177-eNOS levels. This differs from our previous work reporting decreased eNOS expression (after 6 days) [13], and from a study by He et al. who observed reduced eNOS and Ser1177-eNOS levels after 3 weeks of DOX (2.5 mg/kg over 3 weeks; 15 mg/kg cumulative) [15], again raising awareness for the influence of timing and repetitive DOX administration on vascular toxicity.”

Finally, we have acknowledged the potential difference between acute and chronic treatment also in the Limitations part of the Discussion section as follows (from line 307 to 309, p. 14): 

“First, the present study focusses on the acute adverse effects of DOX on the vasculature (i.e. 16h after administration), which may differ from longer-term effects of repetitive dosing.”

• Why weren’t the cell viability experiments also performed in vascular cells? This seems like something that would have been easy to do and would allow for more direct comparisons between effects of DEXRA in the heart vs. vasculature. 

We agree with the reviewer and have performed additional cell viability experiments on endothelial cells (ECs) and vascular smooth muscle cells (VSMCs) as well (cf. supra; comment 6 of Reviewer #1).

• What is the sample size for Top-11beta protein expression? The authors should include a quantification that includes the full sample. Does DOX increase TOP-IIbeta expression? If not, how do you know that the beneficial effect of DEXRA in your experimental setup (in cardiomyocytes) is mediated by depleting TOP-IIbeta?

TOP-IIβ expression was quantified using qPCR and Western blotting in both aortic and cardiac murine tissue on all in vivo samples (cf. supra; comment 5 of Reviewer#1).

• The manuscript seems very under referenced, with a high percentage of the only 11 citations coming from the authors. There are a large number of papers now investigated cardio-toxic effects of DOX and potential interventions to mitigate it that have been omitted.

We apologise for this as the original manuscript was intended to be published as a short communication, which has a maximum of 12 references. Now, the manuscript is a full research article and, where appropriate, we have elaborated more on current literature throughout the manuscript.

• Lines 192-193: This should be tempered. You cannot conclude from your data that DOX impaired endothelial function by influencing eNOS. E.g., DOX could be lowering NO-dependent dilation via greater ROS scavenging.

To further investigate the role of ROS in DOX-induced endothelial dysfunction, aortic segments were treated with either vehicle, DOX or a combination of DOX (1 μM) and the antioxidant NAC (10 μM). NAC did not prevent DOX-induced impairment of ACh-stimulated vasodilation, suggesting that DOX impairs endothelial function independently from ROS-mediated pathways, at least in the currently evaluated acute setting (cf. supra; comment 8 of Reviewer #1).

• Statistical tests used should be described in the “Methods – Statistical Analysis” section.

Information on statistical tests has been added to the Statistical Analysis part of the Materials & Methods section as follows (from line 178 to 182, p. 8): 

“The following statistical tests were performed: For cell viability experiments, a one-way ANOVA with Tukey’s multiple comparisons test; for ACh and DEANO curves, a repeated measures two-way ANOVA with Dunnett’s multiple comparisons test; for PE contraction with and without L-NAME, a one-way ANOVA with Dunnett’s multiple comparisons test; and for RT-qPCR and WB, a Mann-Whitney U test.”

---

## [Editor Report · Decision Letter 1]

9 Nov 2023

Dexrazoxane does not mitigate early vascular toxicity induced by doxorubicin in mice

PONE-D-23-14227R1

Dear Dr. Bosman,

We’re pleased to inform you that your manuscript has been judged scientifically suitable for publication and will be formally accepted for publication once it meets all outstanding technical requirements.

Kind regards,

Peng Zhang, Ph.D.

Academic Editor

PLOS ONE
---

## [Editor Report · Acceptance letter]

16 Nov 2023

PONE-D-23-14227R1 

Dexrazoxane does not mitigate early vascular toxicity induced by doxorubicin in mice 

Dear Dr. Bosman:

I'm pleased to inform you that your manuscript has been deemed suitable for publication in PLOS ONE. Congratulations! Your manuscript is now with our production department. 

Kind regards, 

on behalf of

Prof. Peng Zhang 

Academic Editor

PLOS ONE